# The Importance of Stem Cells Isolated from Human Dental Pulp and Exfoliated Deciduous Teeth as Therapeutic Approach in Nervous System Pathologies

**DOI:** 10.3390/cells12131686

**Published:** 2023-06-22

**Authors:** Niccolò Candelise, Francesca Santilli, Jessica Fabrizi, Daniela Caissutti, Zaira Spinello, Camilla Moliterni, Loreto Lancia, Simona Delle Monache, Vincenzo Mattei, Roberta Misasi

**Affiliations:** 1National Center for Drug Research and Evaluation, Istituto Superiore di Sanità, 00161 Rome, Italy; 2Biomedicine and Advanced Technologies Rieti Center, Sabina Universitas, 02100 Rieti, Italy; f.santilli@sabinauniversitas.it (F.S.); j.fabrizi@sabinauniversitas.it (J.F.); v.mattei@sabinauniversitas.it (V.M.); 3Department Experimental Medicine, “Sapienza” University of Rome, 00161 Rome, Italy; daniela.caissutti@uniroma1.it (D.C.); zaira.spinello@uniroma1.it (Z.S.); roberta.misasi@uniroma1.it (R.M.); 4Department of Neuroscience, Mental Health and Sensory Organs (NESMOS), “Sapienza” University of Rome, 00189 Rome, Italy; camillamoliterni@gmail.com; 5Department of Biotechnological and Applied Clinical Sciences, University of L’Aquila, 67100 L’Aquila, Italy; loreto.lancia1@graduate.univaq.it (L.L.); simona.dellemonache@univaq.it (S.D.M.)

**Keywords:** stem cells, DPSCs, dental pulp stem cells, SHEDs, stem cells from human exfoliated deciduous teeth, neurodegenerative diseases, cell therapy

## Abstract

Despite decades of research, no therapies are available to halt or slow down the course of neuro-degenerative disorders. Most of the drugs developed to fight neurodegeneration are aimed to alleviate symptoms, but none has proven adequate in altering the course of the pathologies. Cell therapy has emerged as an intriguing alternative to the classical pharmacological approach. Cell therapy consists of the transplantation of stem cells that can be obtained from various embryonal and adult tissues. Whereas the former holds notable ethical issue, adult somatic stem cells can be obtained without major concerns. However, most adult stem cells, such as those derived from the bone marrow, are committed toward the mesodermal lineage, and hence need to be reprogrammed to induce the differentiation into the neurons. The discovery of neural crest stem cells in the dental pulp, both in adults’ molar and in baby teeth (dental pulp stem cells and stem cells from human exfoliated deciduous teeth, respectively) prompted researchers to investigate their utility as therapy in nervous system disorders. In this review, we recapitulate the advancements on the application of these stem cells in preclinical models of neurodegenerative diseases, highlighting differences and analogies in their maintenance, differentiation, and potential clinical application.

## 1. Introduction

The increasing median age of the global population renders neurodegeneration and age-related deficiencies a major public health concern worldwide [1]. Neurodegenerative diseases are a heterogeneous group of fatal, progressive nervous system disorders, characterized by the deposition of specific protein aggregates that spread through predictable neuronal routes [1,2,3]. A pathogenic phenotype is overt only after many years of silent pathology, during which malignant protein aggregation and metabolic derangement take place. Despite decades of research and significant advancements in early diagnosis [4,5,6], no current therapeutic strategies are available that could halt or reverse the progression of the diseases. In each neurodegenerative disorder, pharmacological intervention is limited to symptom treatment. For instance, clinically approved drugs for Alzheimer’s Disease (AD) are represented by acetylcholinesterase inhibitors such as donepezil, rivastigmine, and galantamine, aimed to restore early deficits of the subcortical cholinergic system [7]; and N-Methyl-D-Aspartate receptor antagonist (memantine), which prevents glutamate-induced excitotoxicity [8]. Attempts to develop antibody-based therapy targeting either Amyloid-β (Aβ) peptides or Tau, the two major aggregating proteins involved in AD pathogenesis led to conflicting clinical results and failed to show benefits on primary efficacy endpoints [9]. Similarly, no disease-modifying therapies have been developed for other neurodegenerative diseases such as Parkinson’s Disease (PD) and Amyotrophic Lateral Sclerosis (ALS). Indeed, current therapeutic approaches for these disorders aim to alleviate symptoms, such as dopamine precursors (i.e., Levodopa) for PD [10] or anti-glutamatergic drugs (i.e., Riluzole) for ALS [11]. 

Failure to identify an effective therapeutic molecule could be ascribed to: (1) the widespread neuronal death affecting distant neuronal populations; (2) the lack of spontaneous neuronal regenerative potential in the central nervous system (CNS); and (3) the presence of the blood–brain barrier (BBB), which needs to be overcome by the therapeutic agent in order to exert its effect in the CNS. In the present review, we highlight the ability of DPSCs and SHEDs to form different sub-population of neurons and summarize the recent advancements of cell therapy (CT) using DPSCs and SHEDs in pre-clinical models of neurodegenerative disorders and sensory system pathologies.

## 2. Types of Stem Cells for Therapy of Neurodegenerative Diseases

The lack of successful drugs able to significantly affect the onset and the progression of neurodegeneration led researchers to investigate alternative therapeutic approaches, with CT being the most promising one, since it has proven to be efficient against numerous diseases [12,13,14]. CT is a form of regenerative therapy that consists of the administration of stem cells capable of self-renewal and differentiation either through injection or as grafts. Stem cells can either be produced upon reprogramming of a mature cell into an embryonal-like cell (induced pluripotent stem cells, iPSCs), derived from embryonic tissue (ESCs), or from adult mesenchymal somatic stem niches, including the bone marrow (BM-MSCs), the dental pulp, and adult neural stem cells (NSCs) [15,16,17,18]. The latter are found buried deep in subcortical regions, namely the subventricular zone of the lateral ventricles and in the subgranular zone in the dentate gyrus of the hippocampus [19,20]. iPSCs offer the advantage of being highly accessible through a skin or blood biopsy, display low risk of immune rejection since can be autologously transplanted and pose no ethical issues as they are extracted from adult patients. Nonetheless, the extraction of adult cells is still a fairly invasive procedure that can be painful to patients, and the cells carry every genetic or epigenetic modification of the donor [14,21]. On the other hand, ESCs are pluripotent, possessing unlimited proliferative abilities. However, besides the possibility of immune rejection, the main limitation is that blastocysts are extracted from embryos, posing huge ethical concerns [14,22]. MSCs have a limited differentiation potential, as they are multipotent rather than pluripotent, meaning that they are already committed to a specific fate, but display minimal tumorigenic potential compared to ESCs and iPSCs [23,24]. Lastly, although NSCs are direct precursors of neurons, it is not possible to collect human adult neural stem cells, making them not suitable for CT. Among MSCs, those derived from bone marrow are traditionally the most utilized cell line. However, they have been reported to fail to form mature and functional neurons [25] due to their mesodermal lineage, although conflicting results have been reported [26]. Moreover, the procedure for the extraction of BM-MSCs is invasive and painful, making BM-MSCs less than ideal. A more promising source of MSCs for the treatment of CNS pathologies is the one derived from the dental pulp of permanent and exfoliated teeth (DPSCs and SHEDs, respectively) [27,28]. This cellular population offers several advantages compared to other stem cells. Dental pulp soft tissue is easily accessible and not invasive. In the case of SHEDs, once a baby tooth falls out, its collection and storage may be performed [29]. SHEDs are derived from exfoliated teeth, meaning that they can be collected from what is usually considered waste material [30], whereas DPSCs are typically collected from the third molar (the wisdom teeth), normally considered waste material as well and whose extraction is a routinary performed in odontoiatric practice. The dissociation of DPSCs and SHEDs from the soft dental pulp is also easy and their maintenance requires materials normally found in cellular biology laboratories [28,29]. Foremost, DPSCs and SHEDs derive from the neural crest, thus already committed toward the neuronal lineage. For these reasons, DPSCs and SHEDs are gaining increasing interest in CT for neurodegenerative disorders. The different types of stem cells used in CT, their embryonal origin and their terminal commitment are depicted in Figure 1.

## 3. Induction of Neuronal Lineages in DPSCs and SHEDs

A major advantage for the usage of dental stem cells is that, once collected from teeth, they can be maintained in commonly used media [28,29] and, as detailed in Table 1, several media can be used to differentiate specific subpopulations of neuronal cells from DPSCs and/or SHED. Several authors have demonstrated that DPSCs are positive for the typical MSCs markers recognized by International Society for cell and Gene Therapy [31] such as CD73, CD90, and CD105, and negative for the markers CD14, CD19, CD34, CD45, and HLA-DR [12,13]. Although no specific DPSCs markers have yet been identified, the expression of a plethora of neuronal markers has been reported upon induction of the neuronal lineage, including GFAP, beta-III-Tubulin, Notch-2, Synaptophysin, MAP-2, and Nerve Growth Factor (NGF) receptors [31,32,33]. In order to maximize neuronal maturity, serum-free conditions have also been tested [34,35], resulting in a reduced proliferation and avoidance of a possible immune reactivity. Hypoxia was also reported to induce the neuronal differentiation of DPSCs in vitro [36]. Differentiation into various sub-population of neurons has been achieved through different approaches, such as the usage of chemical inductors, growth factors and three-dimensional culturing (neurospheres [37], providing a neuronal niche that allows for a better differentiation of DPSCs). By producing specific neuronal lineages, different brain regions can be modeled, reflecting the areas most affected by neurodegeneration. For instance, glutamatergic neurons are mostly found in the hippocampus and in the neocortex, where neuronal loss is characteristic of late stages of major neurodegenerative disorders [38,39,40]; inhibitory GABAergic neurons are also found in the neocortex and in subcortical areas, such as the medium spiny neurons in the striatum, affected in Huntington Disease (HD). Cholinergic neurons are typically found in the ventral horn of the spinal cord and are compromised in ALS, but are also present in subcortical nuclei in the basal forebrain, one of the first deranged regions in AD [41], whilst Dopaminergic neurons found in the striatum are mainly affected by PD. Using a combination of growth factors and chemical inductors, a pan-neuronal phenotype can be produced. The stimulation of the fibroblast growth factor (FGF/FGFR) pathway activates ERK signaling in DPSCs, promoting neuronal differentiation and increasing the size of neurospheres [42,43]. Functionally active neurons have been produced by the combination of bFGF and a demethylating agent, together with the addition of cyclic adenosine monophosphate (cAMP) and PKC activators such as forskolin and tissue-type plasminogen activator [44]. The glutamatergic and GABAergic lineages were produced by culturing DPSCs in a medium containing bFGF and EGF with the addition of the Peptidyl-prolyl cis-trans isomerase NIMA-interacting 1 (PIN-1), which decreased the number of Dopaminergic neurons and increased the number of Glutamatergic and GABAergic neurons [45]. The Dopaminergic lineage can be produced in vitro by the addition of brain-derived-growth factor (BDNF), glial-derived neurotrophic factor (GDNF), and neurotrophin-3 (NT-3), as shown by the expression of the dopaminergic marker tyrosine hydroxylase (TH) [46]. Lastly, the formation of cholinergic motor neurons from DPSCs was achieved by the pre-induction with sonic hedgehog and RA together with neurotrophins BDNF, GDNF, Insulin-like growth factor (IGF-1), and cAMP activators [47]. DPSCs have also shown the potential to differentiate into pain-sensitive peripheral bipolar neurons [48] by addition to culture medium of Retinoic Acid (RA) and NGF. These neurons resulted to be positive to the marker for peripheral neurons Brn-3a, for the nociceptor TRPV1, and for the pain neurotransmitter substance-P. Moreover, Ca^2+^ imaging demonstrated that these neurons were also active in terms of electrical activity and showed to be responsive to capsaicin in a similar fashion compared to rats’ dorsal root ganglia cultured cells. Together, this evidence suggests that DPSCs may be a candidate to model peripheral, pain-sensitive neurons. As detailed in Table 1, multiple approaches have been reported to successfully produce specific sub-population of neurons from DPSCs and/or SHEDs, suggesting that these cells could represent invaluable models for pathologies, such as neurodegenerative disorders that selectively affect neurons with different neurochemical properties. In the next sections, we will review the latest reports on the application of DPSCs and SHEDs both as models for the study of neurodegeneration and as therapeutic interventions against these devasting conditions.

This section may be divided by subheadings. It should provide a concise and precise description of the experimental results, their interpretation, as well as the experimental conclusions that can be drawn.

## 4. Relevance of DPSCs and SHEDs in Neurodegenerative Diseases

Due to their neural crest origin, the high proliferation rate and their ability to differentiate into functional neurons, DPSCs represent a promising tool, both for basic research and therapeutic strategies to fight neurodegenerative diseases. DPSCs have been reported to express high levels of pluripotent stem cells markers such as Oct4, Nanog, Sox2 and Klf4 [49,50], possess more potent immunosuppressive activity and neurogenicity and display non-tumorigenic properties compared to other MSCs [51,52]. Neuronal-induced DPSCs have been reported to possess voltage-gated channels, making them electrically active cells. Moreover, under the proper conditions, DPSCs have been reported to differentiate into specific sub-populations of neurons such as dopaminergic or cholinergic neurons [47,53]. A key aspect that sets apart DPSCs from other mesenchymal stem cells in the study of neurodegeneration is their enhanced ability to migrate toward a region of neuronal damage. In a recent and elegant work [52], neuronal damage was induced in vitro in hippocampal primary mice neurons by exposure to kainic acid, causing excitotoxicity, a hallmark of AD and other neurodegenerative disorders [54]. Next, the migratory activity of DPSCs and BM-MSCs was addressed following the transwell migratory assay, whilst their ability to penetrate into the extracellular matrix was assessed by matrigel invasion assay. In both experimental setups, DPSCs showed a greater ability to migrate towards a neurodegenerative milieu compared to BM-MSCs, mirrored by a higher expression of homing factors. Interestingly, the conditioned medium from kainate-treated cells was sufficient to drive the migration of DPSCs, highlighting their ability to sense neuronal damage. Together, this result demonstrates that DPSCs are better suited for the study of neurodegeneration compared to other MSCs. In this section, we will summarize the recent advances on the application of DPSCs, SHEDs and their secretome in pre-clinical models of sensory system disorders and neurodegenerative diseases. As a caveat, we will not discuss tissue regeneration or ischemic stroke, as these topics fall outside the aim of this work and have been extensively explored elsewhere [55,56,57,58,59,60].

### 4.1. Potential Therapeutic Application of DPSCs and SHEDs for Sensory System Disorders

The remarkable potential to differentiate into different specialized neurons has led researchers to investigate whether CT with DPSCs or SHEDs could be beneficial against sensory system disorders (i.e., visual and auditory systems). Indeed, as compared to other MSCs, DPSCs show the ability to differentiate into specialized terminal neurons with limited self-renewal properties such as photoreceptors and hair cells. 

Transplant of DPSCs into the eye of a retinal degeneration rat model resulted in the differentiation of DPSCs into retinal cells [61]. Lam and co-workers [62] assessed the potential rescue exerted by the combination of subretinal and systemic injection of DPSCs towards retinal degeneration induced in rats by NaIO_3_, which triggers caspase-dependent apoptosis of the retinal pigment epithelium. Although no significant differences were found in terms of retina histology, researchers showed that the concerted effort of DPSCs administered through two different routes resulted in significant recovery in the photopic 30 Hz flicker ERG response, associated with an improved cone pathway response. 

The ability of DPSCs and SHEDs to differentiate into hair cells was explored by Gonmanee and colleagues [63] by co-culturing them with rats’ auditory brainstem slices. DPSCs or SHEDs were first cultured in a medium supplemented with b-FGF and EGF in low-binding dishes. Neurospheres, formed in this condition, positive to the neuronal precursor marker Nestin, were next transplanted into organotypic auditory brainstem slices. In this setting, DPSCs and SHEDs showed the ability to migrate outside the neurosphere and to differentiate from fibroblast-like to neuron-like morphology. This result demonstrates that auditory brainstem slices offer a microenvironment that promotes survival, expansion and neurogenesis of transplanted stem cells. A similar differentiation approach was adopted in another study, where DPSCs were induced to differentiate into cochlear hair cells [64]. Researchers first induced the neuronal lineage by the addition of b-FGF and EGF for a week. Next, terminal differentiation in hair cells was induced by the administration of EGF and IGF-1 for two more weeks, resulting in the formation of hair cells.

Collectively, the results here presented highlight the ability of DPSCs and SHEDs to differentiate into specialized neurons such as photoreceptors and cochlear hair cells both in vitro and within a specific micro-environment. These properties render both DPSCs and SHEDs very promising lineages for CT of sensory system disorders and, in general, for neuronal damage, as will be discussed in the next sections.

### 4.2. DPSCs in Alzhemier’s Disease Models

AD is the most common form of neurodegeneration, accounting for more than half of the total cases of dementia [65,66]. Despite decades of research, no current strategies are available to slow down or halt the course of the disease, which involves the accumulation of neurotoxic proteins such as Aβ and Tau and neuroinflammation, leading to synaptic loss, neuronal death and brain atrophy. Since the classical pharmacological approach resulted to be ineffective against AD, CT is gaining increasing attention as an alternative to treat AD and similar neurodegenerative disorders [49,67,68]. Amongst many potential candidate cell types, DPSCs appear to be the most promising cell lineage due to their unique characteristics. Indeed, they have been reported to ameliorate Aβ-induced damage in vitro models by releasing neuroprotective factors such as BDNF, GDNF and NGF among other neurotrophins [69]. Similarly, DPSCs have been shown to protect neurons from neuronal degeneration induced by okadaic acid [49,70,71]. The pathology can be replicated in vitro by overexpressing Aβ peptides in neuronal cells such as the commonly used SH-SY5Y neuroblastoma cells [72]. In this model, the secretome derived from DPSCs was shown to increase cell viability and to exert anti-apoptotic effects as observed by the up-regulation of the anti-apoptotic factor Bcl-2 and the down-regulation of the pro-apoptotic factor Bax. This effect was attributed to the higher amount in DPSCs secretome of the protein Neprilysin compared to BM-MSCs, the rate-limiting metalloprotease involved in the degradation of Aβ 1-42 peptide [73]. In another study [74], the protective effect of stem cells on kainate-induced excitotoxicity was assessed in vivo by intrahippocampal injection of kainic acid followed by transplantation of either DPSCs, BM-MSCs or their respective conditioned media. An eight-arm radial maze test was performed to assess spatial memory and learning, revealing that both stem cells and their media significantly improved memory acquisition. Histochemical analyses further showed that hippocampal cytoarchitecture was preserved by stem cells transplant, with DPSCs showing greater protective potential against kainic acid compared to BM-MSCs. Accordingly, neuroinflammation was reduced and neurogenesis was increased both by stem cells and their media. Zhang and co-workers evaluated the therapeutic potential of DPSCs in a rat model of AD constructed by the intrahippocampal injection of human Aβ 1-42 peptide [75]. Transplantation of DPSCs after induction of Aβ toxicity resulted in increased levels of Doublecortin (early neuronal marker, indispensable for neuronal migration), NeuN (nuclear marker of mature neurons), and NF200 (a neurofilament subunit), indicating that DPSCs are integrated in rats’ hippocampus. Radial maze further confirmed the protective effect of DPSCs, displaying increased learning and memory formation compared to non-transplanted rats. Together, these results highlight the ability of both DPSCs and their media to protect neurons against typical neurodegeneration features in vivo.

In addition to Aβ, AD pathology is characterized by intraneuronal inclusion made up of the microtubule-associated protein Tau [76]. Gazarian and colleagues [77] exploited the neuronal features of DPSCs to model a cellular system capable of recapitulating Tau aggregation and pathogenesis. Upon induction of DPSCs neuronal lineage, they revealed the presence of both Tau mRNA and protein. Epitope mapping performed by using different Tau antibodies showed a moderate phosphorylation pattern of Tau, including pathogenic residues Ser404 [78], Ser 422 [79], and Thr231 [80]. However, phosphorylation at residues Thr212/Ser214, a pathogenic hallmark of AD [81], could not be detected within DPSCs cytoplasm. Overall, this study suggests that DPSCs are a suitable system for modeling physiological and pathological chances associated with Tau aggregation, whose biology is at the crossroad of many neurodegenerative diseases.

### 4.3. DPSCs and SHEDs in Parkinso’s Disease Models

PD is the second most common neurodegenerative disorder, affecting about 1% of the global population aged over 65 [82]. It is a progressive disorder causing the loss of dopaminergic neurons within the substantia nigra due to the presence of insoluble aggregates made up of the protein alpha-synuclein, causing motor deficits such as bradykinesia, resting tremors and rigidity. Similar to AD, no therapeutic strategies are currently available to halt the progression of this disease [83]. Being able to differentiate into dopaminergic neurons and release neurotrophic factors, DPSCs represent a promising treatment for PD and other related synucleinopathies [84].

In order to evaluate the therapeutic potential of DPSCs towards PD, Simon and co-workers [85] induced parkinsonian symptoms in a mouse line by administration of 1-methyl-4-phenyl-1,2,3,6-tetrahydropyridine (MPTP, a classical treatment to produce PD phenotype in rodents [86]). DPSCs were first differentiated into mature dopaminergic neurons, as reported by the down-regulation of pluripotent markers Oct4, Sox2, and Nestin, whilst mature neuronal markers such as beta-Tubulin, TH, Dopamine Transporter, and MAP2 were up-regulated. After the intranasal application of DPSCs, both motor and olfactory functions were recovered in MPTP-treated mice, suggesting that DPSCs could attenuate PD pathology. Importantly, as shown by immunofluorescence staining after labeling DPSCs with the bio-orthogonal compound PKH26, DPSCs are able to integrate within the damaged neuronal area and can be detected even after four weeks, strongly indicating that the intranasal route of administration could be an exploitable strategy for future therapies. This route of administration appears to be ideal, as it is not invasive and bypasses the BBB. In another in vivo study [87], PD pathology was induced in a rat model by the administration of 6-hydroxydopamine (6-OHDA). Next, SHEDs differentiated into dopaminergic neurons were transplanted, resulting in reduced neuronal damage.

DPSCs are expected to become central in treating neuropathological disorders in the future. Some pre-clinical evidence on other neurodegenerative disorders has been produced, as recapitulated in the next section. Moreover, clinical trials have also been registered and are discussed in the last section.

### 4.4. Application of DPSCs in Other Neurodegenerative Disorders

In addition to AD and PD, the usefulness of DPSCs as a neuroprotective therapeutic approach against neurodegenerative diseases has been assessed in other diseases and models. Table 2 summarizes the preclinical models of neurodegeneration in which DPSCs and/or SHEDs have been tested for their therapeutic potential. Aliaghaei et al. [88] investigated the neuro-restorative potential of DPSCs in a model of Cerebellar Ataxia (CA), a common pathological feature in neurodegeneration affecting the cerebellum. CA was induced in rats by the administration of the neurotoxin 3-acetylpyridine (3-AP), followed by bilateral cerebellar transplantation of DPSCs. Behavioral testing revealed an increase in motor skills and muscular activity, along with the preservation of cerebellar physical parameters such as volume and layers’ cytoarchitecture. Purkinje cells, mostly affected by CA, were protected by DPSCs grafting, and the release of pro-inflammatory cytokines was reduced, pointing toward a neuroprotective and restorative effect of DPSCs towards CA induced by 3-AP.

In a rat model of vascular dementia (VaD, the second most common form of dementia [89]), Zhang and co-workers evaluated the therapeutic potential of DPSCs [90]. The pathology was induced by the widely used hypoperfusion approach by occluding two vessels. Next, DPSCs labeled with the PKH67 dye were injected into the tail vein. Using this approach, researchers were able to identify injected cells in the recipient animal and demonstrated that DPSCs can migrate toward the injury site, where they differentiate into mature neurons as shown by the increase in neuronal markers Doublecortin, NF200, and NeuN. These results were paralleled by the behavior test using the eight-arm maze, showing that both latency and error rate was reduced after injection of DPSCs.

DPSCs therapeutic potential has been investigated in models of HD as well, an autosomal dominant progressive neurodegenerative disorder caused by the expansion of CAG codon on the short arm of chromosome 4, coding for the protein huntingtin [91]. Whereas healthy individuals have 6–35 repeats of this codon, individuals with HD possess more than 36 repeats and up to 120 repeats [92]. The expansion of this codon leads to synaptic loss and trafficking deregulation, resulting in the loss of GABAergic medium spiny neurons in the striatum. Akin to all other neurodegenerative diseases, no treatment is currently available to modify and halt the progression of the pathology. Eskandari and colleagues [93] were the first to probe the therapeutic potential of DPSCs in a rat model of HD. The pathology was induced by the administration of 3-nitropropionic acid (3-NP), an established protocol to mimic HD in murine models [94], which inhibits the respiratory chain complex II, resulting in neuronal loss and movement disorders. DPSCs were bilaterally transplanted in the medio-posterior part of the striatum two days after 3-NP injection. Whereas 3-NP injected animals displayed reduced dendrites length of medium spiny neurons, reduced motor skills and muscular activity, DPSCs grafting rescued both behavioral and histological abnormalities caused by 3-NP, increasing cells’ survival and blocking astrogliosis and microgliosis. Moreover, cleaved Caspase-3 expression, which was dramatically increased by 3-NP, returned to levels comparable to untreated mice, indicating a reduction in apoptosis following DPSCs transplant. Similarly, mRNA levels pro-inflammatory cytokines were reduced by DPSCs grafting, suggesting an overall rescue of the pathogenic phenotype triggered by 3-NP. The same HD model using 3-NP in rats was adopted by Wenceslau and co-workers [95]. SHEDs were administered via intravenous injection in order to investigate their ability to cross the BBB and to engraft into the damaged brain region. Confocal microscope imaging showed that SHEDs could reach the striatum, hippocampus and cortex, where they differentiated into pericyte-like and neuronal-like cells. Moreover, 3-NP-treated, DPSCs-infused rats expressed BDNF levels comparable to those of untreated animals, whereas 3-NP-treated mice displayed minimal BDNF expression. Importantly, striatum-homed cells expressed high levels of Dopamine receptor 2 and DARPP32, a specific marker for medium spiny neurons, suggesting that SHEDs transplant could induce neuronal regeneration.

Lastly, DPSCs conditioned media (DPSCs-CM) was assessed for its therapeutic potential also in a model of ALS [91]. ALS is the main motor neuron disease, characterized by the progressive loss of upper and lower motor neurons [96]. Here, researchers employed the common ALS murine model TgSOD1^G93A^, modeled after the mutation on the protein Copper/Zinc Superoxide Dismutase-1 (SOD1), associated with familial cases of ALS [97,98]. Animals at the early presymptomatic stage (post-natal day 35 to 47) were treated with DPSCs-CM through intraperitoneal route and observed that DPSCs-CM ameliorated the neuromuscular junction denervation compared to vehicle-treated controls. Similar results were obtained when DPSCs-CM was administered at the late pre-symptomatic stage (post-natal day 91), together with a reduced motor neuron loss at the ventral horn of the spinal cord. However, astrocytes and microglia reactivity in the spinal cord were unaffected by DPSCs-CM injection. Nevertheless, when mice were treated from symptoms onset until end-stage, the lifespan of DPSCs-CM-treated animals significantly increased compared to control animals.

**Table 2 cells-12-01686-t002:** Usefulness of DPSCs and/or SHEDs in pre-clinical models of neurodegenerative diseases.

Pathology	Pre-Clinical Model	DPSC Administration	Outcome	References
Alzheimer’s Disease	Primary rat hippocampal cultures treated with Amyloid-β 1–42 (5–10 µM) or 6-OHDA (5–40 µM) for 24 h	Co-culture with primary neurons	Rescue of cell viability; increase in expression of neuronal markers: release of neurotrophins	[69]
	Human neuroblastoma SH-SY5Y cells treated with 20 nmol/L Okadaic Acid for 24 h	Transwell insert with porous membrane	Restoration of morphology and cell viability; reduction in apoptosis; reduction in phospho-Tau	[70]
	Human neuroblastoma SH-SY5Y cells treated with 5 µM Amyloid-β 1–42	Transplantation of DPSCs secretome	Increased cell viability; up-regulation of anti-apoptotic Bcl-2; down-regulation of pro-apoptotic Bax	[49]
	Mice treated with Kainic Acid	Intrahippocampal transplantation of DPSCs or their secretome	Reduction in cognitive impairment; improved memory acquisition; reduction in neuroinflammation; increase in neurogenesis	[74]
	Rats treated with 1 mg/mL Amyloid-β 1–42	Intrahippocampal transplantation of DPSCs	Increased secretion of neurotrophins; improved cognitive behavior	[75]
Parkinson’s Disease	Rats intraperitoneally injected with 20 mg/kg MPTP	Intranasal	Increased senosory-motor coordination; rescue of olfactory functions; increase in TH-positive neurons	[85]
	Rats injected unilaterally in the striatum with 10 µg/µL 6-OHDA	Injection of SHEDs in the striatum	Recovery of neurological behavior; increased survival; increase in TH-positive neurons	[87]
Cerebellar Ataxia	Rats injected intraperitoneally with 75 mg/kg 3-Acetylpyridine	Intracerebellar injection	Enhanced motor skills; enhanced muscle activity; rescue of cerebelar volume; reduction in inflammatory cytokines	[88]
Vascular Dementia	Two-bessel occlusion in rats	Injection of marked murine DPSCs into tail veins	Successful migration of DPSCs into the lesioned areas observed by PKH compounds; increased neuronal markers; improved behavioral performances	[90]
Huntington’s Disease	Rats injected intraperitoneally with 30 mg/kg of 3-nitropropionic acid	Bilateral transplantation of marked DPSCs	Improved motor skills and muscle activity; increased neurite length; reduced astrogliosis and microgliosis; downregulation of Caspase-3 activity; decreased expression of inflammatory cytokines.	[93]
	Rats injected intraperitoneally with 20 mg/kg of 3-nitropropionic acid	Intravenous injection of SHEDs	SHEDs can cross the BBB; increased expression of neurotrophins;	[95]
Amyotrophic Lateral Sclerosis	Tg-SOD1^G93A^ mouse model	Transplantation of DPSCs secretome	Reduced neuromuscular junction denervation; reduced muscle atrophy; reduced neuronal loss; extended lifespan.	[96]

## 5. DPSCs- and SHEDs-Based Clinical Trials for Neuropathological Disorders

In the last decade, several clinical trials utilizing either DPSCs or SHEDs have been registered in ClinicalTrials.gov and in the International Clinical Trial Registry Platform (ICTRP). Unsurprisingly, most of them deal with odontoiatric disorders such as pulp necrosis and periodontitis. However, DPSCs clinical application extends beyond the oral cavity, with clinical trials registered for treatment of COVID-19 [99,100], acute ischemic stroke [101], osteoarthritis [102], Type 1 diabetes [103], and systemic lupus erythematosus [104]. Notably, three clinical trials have been registered for the treatment of HD [105,106,107], submitted by the company Azidus Brasil. The first Phase I clinical trial [105] started in 2016 and aimed to evaluate the effect of stem cell of the therapeutic formulation Cellavita HD. Participants with HD received either a low dose (1 × 10^6^ cells/weight range, n = 3) or a high dose (2 × 10^6^ cells/weight range, n = 3) of Cellavita HD through three intravenous injection, one every 30 days. No results have been posted at the moment, as the study is expected to end by 23 December 2023. The company registered a Phase II clinical study [106] designed as a prospective, monocentric, randomized, triple-blinded placebo-controlled study using two doses of Cellavita HD product. The 35 recruited patients will receive three intravenous administrations for three months, with low dose or high dose of Cellavita HD (similarly to the previous trial) and will be evaluated in terms of clinical neurological symptoms, magnetic resonance imaging, proton spectroscopy and evaluation of biological markers. No results for this clinical trial have been posted on ClinicalTrials.gov. The last clinical trial submitted is a Phase II and Phase III defined as open label, single treatment, extension study for long-term safety and efficacy evaluation of Cellavita-HD intravenous administration [107]. Subjects will receive the highest dose tested in the previous trial for 2 years, in order to evaluate long-term effects of the treatment. Clinical scores and CNS imaging will be used to evaluate the effectiveness of the therapy. Although thus far no clinical trials have been carried out involving other neurodegenerative diseases patients, the ongoing trials on HD (a genetic degeneration diagnosed more accurately compared to other neurodegernerative disorders) could potentially provide invaluable information on the effect of stem cell therapy on the degenerating brain, thus paving the way for a broader application of this therapeutic approach. 

Although no adverse effects from DPSCs and SHEDs transplant were reported in published clinical articles and trials [108], limitation and long-term health concerns still exist, primarily focused on non-directional differentiation (the differentiation toward and unwanted lineage) and potential acceleration of tumor progression. This latter aspect is still controversial, as DPSCs have been reported to both promote [109] and mitigate [110] tumor growth. Lastly, procedural limitation must be taken into account, from the collection of the tissue, which may require additional and painful surgery, to the costly and time-consuming ex vivo expansion in laboratory setting to produce an adequate number of cells and/or CM for therapeutic intervention, which often reduces cells’ self-renewal potential and proliferation abilities [111].

## 6. Conclusions

Overall, DPSCs and SHEDs appear to be the ideal cell lineage for the treatment of neurodegenerative disorders for various reasons: (1) their neural crest embryonic origin confers the ability to differentiate into different types of mature neurons; (2) their extraction is easier compared to other mesenchymal stem cells; (3) autologous transplant allows the circumvention of ethical issues and drastically reduces the risk of rejection; (4) they have been proven to ameliorate various forms of neurodegenerative disorders; (5) they display the ability to cross biological barriers and migrate towards the affected brain region, where they can differentiate in situ. SHEDs may possess an even greater therapeutic potential, as they display higher neurogenic potential and derive directly from waste material [112]. Collection of DPSCs and SHEDs could potentially allow for the generation of stem cells biobanks that could store self-renewable cells for future autologous grafts, thus dramatically reducing transplant rejection.

## Figures and Tables

**Figure 1 cells-12-01686-f001:**
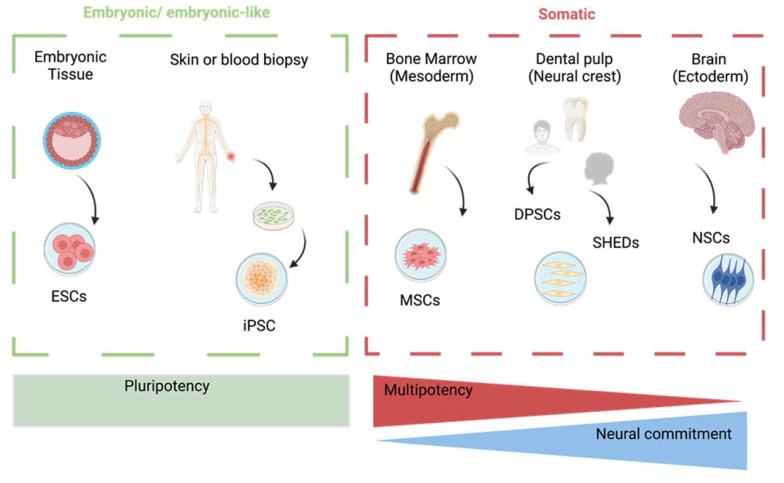
Schematic representation of the main types of stem cells and their embryonic origin.

**Table 1 cells-12-01686-t001:** Protocols for the induction of neuronal lineage from DPSCs and/or SHEDs.

Type and Source of Stem Cells	Medium for Neuronal Lineage	Neuronal Markers	Neuronal Sub-Population	References
Adult DPSCs from third molar	Neurobasal + 2% B27 + 2 mM Gln + 20 ng/mL basic FGF + 20 ng/mL EGF	β3-Tubulin; PLCγ activity	Pan-neuronal	[43]
Adult DPSCs from third molar	DMEM/F12 (1:1) + 2.5% FCS + 10 μM 5-azacytidine + 10 ng/mL basic FGF. After 48 h: 250 μM IBMX + 50 μM forskolin + 200 nM TPA + 1 mM dbcAMP + 10 ng/mL bFGF + 10 ng/mL NGF + 30 ng/mL NT-3 + 1% of insulin-transferrin-sodium selenite premix.	N-tub; NeuN; Neurofilament-M. Electrical activity	Pan-neuronal	[44]
Adult DPSCs from premolar teeth	Neuronal medium + N2 + 20 ng/mL EGF + 20 ng/mL FGF; PIN1 inhibitor (juglone) or PIN1 overepression (through adenovirus)	NeurN; Nestin; VGluT1; GABA; TH	Pan-neuronal; GABAergic; Glutamatergic	[45]
SHEDs from deciduous baby teeth; DPSCs from adult third molar	Neurobasal + 0.5% B27 + 200 ng/mL SHH + 100 ng/mL FGF8 + 50 ng/mL basic FGF + BDNF for 72 h	Nurr1; Engrailed1; Pitx3; Nestin; β3-Tubulin; TH; Ca^2+^ influx	Dopaminergic Neurons	[46]
Adult DPSCs from third molar	**For cholinergic neurons:** DMEM:F12 (1:1) + 1% N2 + 1% non-essential aminoacids + 0.2% Heparin + 0.1 µM RA. After 96 hours: + 100 ng/mL SHH. After 48 h: + 1 µM cAMP 200 ng/mL ascorbic acid. After 72 h: + 10 ng/mL BDNF + 10 ng/mL GDNF + 10 ng/mL IGF-1. **For dopaminergic neurons:** DMEM:F12 (1:1) + 1% N2 + 300 ng/mL Noggin. After 96 h: + 50 ng/mL BDNF + 200 mM Ascorbic acid + 50 µg/mL SHH + 50 µg/mL FGF8b. After 120 h:—bFGF After 72 h: + 10 ng/mL GDNF + 2 µg/mL TGF-βIII + 200 mM cAMP.	Nestin; β3-Tubulin; NeuN; TH; Choline AcetylTransferase	Dopaminergic and Cholinergic Neurons	[47]
Adult DPSCs from third molar	DMEM:F12 (1:1) + 5% FBS + 10 µM non-essential amino acids + 2 mM Glutamatec+ 10 mM RA + 50 µM Ascorbic Acid + 5 µM Insulin + 10 nM Dexamethasone + 20 nM Progesterone + 20 nM Estradiol + 50 ng/mL NGF + 10 ng/mL Thyroxine	Nestin; β3-Tubulin; Brn-3a; TRPV1; substance-P; Ca^2+^ imaging	Peripheral neuronal cells (pain receptors)	[48]

Every media contained antibiotics.

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
