# Peer review of "The Importance of Stem Cells Isolated from Human Dental Pulp and Exfoliated Deciduous Teeth as Therapeutic Approach in Nervous System Pathologies"

_cells, 2023, doi:10.3390/cells12131686_

Round 1

Reviewer 1 Report

The manuscript entitled, “The importance of stem cells isolated from human dental pulp and exfoliated deciduous teeth as therapeutic approach in nervous system pathologies.” Where they have gathered information related to therapeutic approaches using dental pulp derived stem cells. The review is well-designed and properly written; however, the authors needs to clarify my below concerns

1.      As far my information, DPSCs has been used in numerous clinical trials. Please add data from clinicaltrials.org about all the clinical trials registered using DPSCs against neuropathological disorders.

2.      I have found some important work done on using DPSCs for nerve regeneration; however, you have missed such work to cite in your article

i.e. https://pubmed.ncbi.nlm.nih.gov/28657463/, this paper shows that DPSCs can counter balance supra spinal neuro-inflammation, please cite in your article.

Minor English corrections required

Author Response

Please find the reply to Reviewers’ comments highlighted in italics.

The manuscript entitled, “The importance of stem cells isolated from human dental pulp and exfoliated deciduous teeth as therapeutic approach in nervous system pathologies.” Where they have gathered information related to therapeutic approaches using dental pulp derived stem cells. The review is well-designed and properly written; however, the authors needs to clarify my below concerns

We thank the reviewer for appreciating our manuscript.

1.  As far my information, DPSCs has been used in numerous clinical trials. Please add data from clinicaltrials.org about all the clinical trials registered using DPSCs against neuropathological disorders.

We thank the reviewer for raising this major issue. We modified the part related to clinical trials (at the end of section 4.3) and added a novel paragraph (5) to specifically address the revelant clinical trials using DPSC or SHEDs for treatment of neuropathological disorders.

2.      I have found some important work done on using DPSCs for nerve regeneration; however, you have missed such work to cite in your article

i.e. https://pubmed.ncbi.nlm.nih.gov/28657463/, this paper shows that DPSCs can counter balance supra spinal neuro-inflammation, please cite in your article.

Thank you for indicating this interesting work. We added the suggested paper as reference 56.

Reviewer 2 Report

Please, revise the formatting of the summary; it has several words with unnecessary syllable separation.

In the second paragraph of the introduction, review statement 2: the lack of neuronal regenerative potential in the central nervous system (CNS). We know a limited degree of CNS self-repair exists early in development; however, the ability to spontaneously regenerate is dramatically reduced after parturition. The possibility is quite slim, but it exists in some cases. New evidence, provided by single-cell expression profiling, suggests that, following injury, mammalian central and peripheral neurons can revert to an embryonic-like growth state permissive for axon regeneration. Therefore, it would be more appropriate to state: The absence of spontaneous neuronal regenerative potential in the CNS.

The review is well-written and up-to-date. However, we have many reviews on the subject already published. For the manuscript to be more relevant, clinical trials in progress could be discussed more, even if not used directly in neurology. To date, there have been 21 clinical trials registered on ClinicalTrials.gov evaluating the use of DSCs in treating periodontitis post-extraction sockets, edentulous alveolar ridge, cleft lip and palate, knee osteoarthritis, dental pulp necrosis, liver cirrhosis, type 1 diabetes, acute ischemic stroke, post-stroke disability, Huntington’s disease, and COVID-19. In addition to the six studies reported in ClinicalTrials.gov, seven clinical trials were registered on the ICTRP using DSCs to treat periodontitis, wrinkles, and hair loss.

In the same way, it could be argued about the limitations in using these cells and, mainly, the difficulties in standardizing, at the clinical level, the isolation, expansion, and possible differentiation of these cells or their derivatives as exosomes or conditioned medium.

Author Response

Please find the reply to Reviewers’ comments highlighted in italics.

Please, revise the formatting of the summary; it has several words with unnecessary syllable separation.

We thank the reviewer for noticing this formatting error. It has been corrected.

In the second paragraph of the introduction, review statement 2: the lack of neuronal regenerative potential in the central nervous system (CNS). We know a limited degree of CNS self-repair exists early in development; however, the ability to spontaneously regenerate is dramatically reduced after parturition. The possibility is quite slim, but it exists in some cases. New evidence, provided by single-cell expression profiling, suggests that, following injury, mammalian central and peripheral neurons can revert to an embryonic-like growth state permissive for axon regeneration. Therefore, it would be more appropriate to state: The absence of spontaneous neuronal regenerative potential in the CNS.

We thank the reviewer for the comment. We followed the reccomandation by adding the term “spontaneous” to the related sentence. We think this will improve the accuracy of our manuscript.

The review is well-written and up-to-date. However, we have many reviews on the subject already published. For the manuscript to be more relevant, clinical trials in progress could be discussed more, even if not used directly in neurology. To date, there have been 21 clinical trials registered on ClinicalTrials.gov evaluating the use of DSCs in treating periodontitis post-extraction sockets, edentulous alveolar ridge, cleft lip and palate, knee osteoarthritis, dental pulp necrosis, liver cirrhosis, type 1 diabetes, acute ischemic stroke, post-stroke disability, Huntington’s disease, and COVID-19. In addition to the six studies reported in ClinicalTrials.gov, seven clinical trials were registered on the ICTRP using DSCs to treat periodontitis, wrinkles, and hair loss.

We thank the reviewer for the suggestion. We have added a novel paragraph 5 to specifically address the revelant clinical trials using DPSCs or SHEDs for treatment of neuropathological disorders

In the same way, it could be argued about the limitations in using these cells and, mainly, the difficulties in standardizing, at the clinical level, the isolation, expansion, and possible differentiation of these cells or their derivatives as exosomes or conditioned medium.

Limitations to the usage of DPSCs and SHEDs in clinical trials have been added as well at the end of the new paragraph 5